# Acute Downregulation of Zinc α2-Glycoprotein: Evidence from Human and HepG2 Cell Studies

**DOI:** 10.3390/ijms26125438

**Published:** 2025-06-06

**Authors:** Èlia Navarro-Masip, David M. Selva, Cristina Hernández, Andreea Ciudin, Blanca Salinas-Roca, Julia Cabrera-Serra, Rafael Simó, Albert Lecube

**Affiliations:** 1Obesity, Diabetes and Metabolism (ODIM) Research Group, Lleida Biomedical Research Institute (IRBLLeida), University of Lleida, 25198 Lleida, Spain; elianavarromasip@gmail.com (È.N.-M.); blanca.salinasroca@udl.cat (B.S.-R.); 2Diabetes and Metabolism Research Unit, Institut de Recerca Hospital Universitari Vall d’Hebron, Universitat Autònoma de Barcelona, 08035 Barcelona, Spain; david.martinez.selva@vhir.org (D.M.S.); cristina.hernandez@vhir.org (C.H.); andmih@gmail.com (A.C.); juliacabrera8@gmail.com (J.C.-S.); 3Centro de Investigación en Red en Diabetes y Enfermedades Metabólicas (CIBERdem), Instituto de Salud Carlos III (ISCIII), 28029 Madrid, Spain

**Keywords:** ZAG, adipose tissue, insulin, standard meal, glucose tolerance test, glucagon, HepG2 cells

## Abstract

Zinc-alpha2-glycoprotein (ZAG) is a soluble glycoprotein primarily produced in adipocytes and the liver, with key roles in lipid metabolism, including lipolysis and the browning of adipose tissue. Despite extensive studies on its role in rodents, the relationship between ZAG and insulin in humans remains unclear. Given the emerging interest in ZAG’s involvement in metabolic diseases such as metabolic-dysfunction-associated steatotic liver disease, this study aimed to investigate the acute effects of insulin on ZAG levels both in vivo and in vitro. We recruited 24 healthy, individuals who were non-obese and assessed the impact of oral glucose overload, a standardized liquid nutritional supplement, and intravenous glucagon on circulating ZAG levels. In parallel, we explored the effects of insulin on ZAG production in cultured HepG2 cells. Our findings revealed a consistent acute reduction in serum ZAG levels following all in vivo tests, coinciding with increased insulin levels. In vitro, insulin rapidly downregulated ZAG protein and mRNA levels in HepG2 cells, with significant reductions observed within 15 min, followed by partial recovery after 2 h. These results suggest a potential acute inhibitory effect of insulin on ZAG production, supporting its role in promoting energy storage by suppressing lipolysis postprandially. This study provides new insights into the complex interplay between insulin and ZAG in regulating energy balance and highlights the potential of ZAG as a therapeutic target in metabolic diseases.

## 1. Introduction

Zinc-alpha2-glycoprotein (ZAG) is a soluble glycoprotein with a molecular weight of 41 kDa, characterized by a crystal structure that closely resembles that of a class I major histocompatibility complex [1]. Its name derives from its distinctive properties: it precipitates in the presence of zinc and exhibits electrophoretic migration toward the alpha-2-globulin region. ZAG is ubiquitously expressed across various human tissues including the liver, though its primary production occurs in mature adipocytes, particularly within both visceral and subcutaneous fat depots [2,3].

A wide range of biological functions have been attributed to ZAG, including anti-inflammatory activity, involvement in immune responses, ribonucleolytic activity, and the inhibition of tumor proliferation [4,5,6]. Additionally, ZAG has been implicated in the pathogenesis of neurocognitive disorders and conditions such as proliferative diabetic retinopathy [7,8]. However, ZAG’s most prominent role appears in lipid metabolism, where it exerts lipolytic effects, reducing lipogenesis and adipogenesis, while promoting the browning and thermogenesis of adipose tissue [1,9,10]. Supporting this, ZAG is notably overproduced in certain malignant tumors and has been recognized as a lipid-mobilizing factor, playing a critical role in the development of cancer cachexia [11,12,13].

Our understanding of the relationship between ZAG and obesity, as well as its effects on adipose tissue, primarily stems from studies conducted in rodents and in vitro assays [14,15,16,17]. Research involving ZAG-deficient mice demonstrated that ZAG promoted lipolysis in adipocytes and contributed to body weight regulation [15]. Furthermore, the administration of ZAG to mice has been shown to induce a significant, time-dependent reduction in body weight, without affecting lean mass [9,12,18].

In individuals with severe obesity awaiting bariatric surgery, our research group has observed a significant downregulation of ZAG mRNA levels in both subcutaneous and visceral adipose tissues, as well as in circulating levels [3]. Similarly, in individuals with less-severe obesity, ZAG levels were also found to be lower in cases of central obesity and metabolic syndrome, suggesting a negative correlation with adiposity [18,19]. These findings indicate that insulin may play an inhibitory role in regulating ZAG expression, positioning ZAG not only as a key player in the pathogenesis of obesity and metabolic syndrome but also as a potential therapeutic target [12,18]. In addition to its primary production in adipose tissue, ZAG is also expressed in the liver, where it may influence key aspects of hepatic lipid metabolism, including lipogenesis, fatty acid oxidation, and very-low-density lipoprotein (VLDL) secretion [3,12,20]. However, our earlier experiments did not show any significant reduction in ZAG production after culturing HepG2 cells with insulin for five days.

While previous studies have examined ZAG levels in various metabolic conditions, most have focused on static measurements [19,21,22]. Consequently, the impact of acute oral feeding on the circulating ZAG levels in humans remains largely unexplored, as does the effect of acute insulin and incretins on ZAG production in cultured cells. To address this gap, the aim of this study was to investigate the dynamic response of circulating ZAG to three acute stimuli (oral glucose overload, a standardized liquid nutritional supplement, and intravenous glucagon) in individuals without obesity and diabetes to reveal its metabolic adaptations to regulate lipid and energy homeostasis. Understanding these acute regulatory mechanisms may offer a more comprehensive view of ZAG’s physiological role in lipid mobilization and metabolic homeostasis in response to energy intake. Additionally, we sought to examine the acute in vitro effects of insulin on ZAG production.

## 2. Results

### 2.1. Clinical and Biochemical Features

The clinical and biochemical characteristics of the patients included in this study are shown in Table 1. Briefly, 50% of the patients were women, the mean BMI was 24.8 ± 4.9 kg/m^2^, and the fasting glucose levels were 100.4 ± 10.8 mg/dL.

### 2.2. ZAG Serum Levels After Three Insulin Stimulatory Dynamic Tests

The ZAG responses to the three dynamic tests performed are shown in Table 2. During the oral glucose tolerance test (OGTT), a progressive decline in ZAG levels was observed, dropping from 42.1 ± 18.3 µg/mL at baseline to a minimum of 35.2 ± 19.7 µg/mL at 90 min (*p* < 0.05), which coincided with a peak in insulin levels at the same time point (from 13.3 ± 7.7 to 118.3 ± 80.7 pmol/L, *p* < 0.001). In the standardized liquid nutritional supplement test, the ZAG levels also decreased significantly from 42.3 ± 20.3 µg/mL at baseline to 31.5 ± 14.9 µg/mL at 120 min (*p* < 0.05), followed by a slight recovery by 180 min. The insulin levels initially rose, peaking at 60 min (68.2 ± 39.6 pmol/L, *p* < 0.001), before gradually returning closer to baseline by 180 min. Finally, after the intravenous administration of 1 mg of glucagon, the insulin levels surged rapidly, reaching their peak at 5 min (74.4 ± 34.8 pmol/L, *p* < 0.001), while the ZAG levels significantly dropped from 42.2 ± 26.7 µg/mL at baseline to 34.4 ± 17.0 µg/mL at 5 min (*p* < 0.05), with a partial recovery observed by 15 min. This postprandial reduction in ZAG levels, which coincided with increased insulin secretion, suggests a physiological mechanism that transiently suppresses lipolysis to facilitate energy storage.

### 2.3. Effects of Acute Insulin Treatment on ZAG Production in HepG2 Cells

We investigated the impact of insulin (20 µIU/mL) on the ZAG production of HepG2 cells. Both control and insulin-treated HepG2 cells were cultured for 2 h, with media and cells collected at various time points: 0, 15, 30, 60, and 120 min. In vehicle-treated cells, ZAG protein accumulated in the media over time. However, insulin treatment led to a reduction in the ZAG levels in the media as early as 15 min, and these levels remained low for up to an hour before increasing again (Figure 1A). Additionally, the ZAG protein levels within the HepG2 cells were lower at 15, 30, and 60 min compared to those in the vehicle-treated cells over the 2 h period (Figure 1B). Similarly, the ZAG mRNA levels were also reduced by insulin treatment at 15, 30, and 60 min compared to untreated cells, with a recovery in mRNA levels observed at 2 h (Figure 1C).

## 3. Discussion

Overall, our results suggest a consistent acute reduction in serum ZAG levels across patients who underwent three insulin stimulatory dynamic tests (ISDTs). Given that a rapid and temporary increase in plasma insulin levels is a common feature of these ISDTs, our findings support the role of insulin and glucose homeostasis in the immediate regulation of the ZAG response following feeding. This is in line with recent studies that have further elucidated insulin’s role in modulating adipokine secretion, highlighting its impact on systemic metabolic regulation [23]. Our in vivo results confirm that insulin acutely downregulates ZAG expression and production. This is supported by a similar pattern observed in HepG2 cells following insulin administration. Specifically, the ZAG levels in the media of the HepG2 cells were reduced to nearly undetectable levels within 1 h of insulin administration, with subsequent upregulation observed 2 h post-treatment. Additionally, the ZAG mRNA levels in HepG2 cells were significantly reduced within 15 min of insulin treatment and remained low for up to 1 h, with upregulation occurring again after 2 h.

Recent studies have measured basal ZAG levels in different cohorts, including individuals with metabolic syndrome and insulin resistance [19,21,22]. However, these studies do not address how ZAG responds acutely to metabolic changes. Our findings suggest that postprandial insulin secretion downregulates ZAG, aligning with a physiological adaptation favoring energy storage. In this context, the reduction in circulating ZAG levels observed during the glucagon stimulation test was most likely driven by the sharp insulin peak induced by glucagon, rather than by glucagon itself. While in vitro assays using glucagon could further clarify its specific role, our study was designed to explore the integrated metabolic responses to acute stimuli in vivo. This dynamic regulation of ZAG adds a new layer to our understanding of its role in metabolism and could have implications for metabolic disease research. These findings do not contradict our previous study, in which chronic insulin treatment over five days did not alter ZAG production in HepG2 cells [3]. In that study, gene expression was assessed 24 h after the last insulin exposure, likely missing rapid and transient transcriptional changes. In contrast, our current approach focused on short-term (15–120 min) responses, revealing early regulatory mechanisms.

Although the physiological effects of ZAG are primarily related to lipid metabolism, some studies have demonstrated that ZAG also influences glucose metabolism and is linked to insulin resistance. In addition to its well-known role in adipose tissue, ZAG is also produced in the liver, as previously demonstrated by our group and others [3,12]. Hepatic ZAG may participate in the systemic regulation of energy homeostasis by hepatic lipid metabolism modulating, insulin signaling, and possibly glucose handling. Our current findings suggest that insulin directly suppresses hepatic ZAG production, which could contribute to the metabolic shift toward energy storage by means of fat accumulation after nutrient intake.

A systematic review of the current literature investigating the associations between ZAG and dysglycemia found that the circulating ZAG levels were lower in individuals with dysglycemia compared to metabolically healthy controls [24]. Interestingly, Tan et al. found significantly lower serum levels of ZAG and adiponectin in 118 women with components of metabolic syndrome compared to 78 healthy women [25]. However, an examination of the relationship between ZAG and insulin resistance using the euglycemic-hyperinsulinemic clamp, a method involving a controlled and rapid infusion of insulin to maintain constant blood glucose levels, yielded results that were surprising and discordant with ours. In response to hyperinsulinemia, the serum ZAG levels increased significantly in both healthy women and those with metabolic syndrome. The fact that serum insulin levels increased during the clamp from 8.0 ± 3.1 µmol/L to 103 ± 22 µmol/L in healthy women, a concentration nearly 1000 times higher than the peak of 118 pmol/L of insulin achieved during the OGTT in our study, suggests that a short-term, high-intensity insulin signal can temporarily boost ZAG levels.

ZAG has been shown to play an undeniable role in lipid metabolism, such as inducing lipolysis and promoting the browning of white adipose tissue [26,27]. This lipid utilization and mobilization are associated with the downregulation of the transcription factors responsible for lipid accumulation, such as fatty acid synthase, acetyl-CoA carboxylase, and diglyceride acyltransferase, as well as the upregulation of hormone-sensitive lipase, which is involved in lipid transport [20]. Moreover, ZAG mRNA expression has been positively correlated with browning-related genes in subcutaneous white adipose tissue from patients with overweight and obesity, including uncoupling protein 1 and peroxisome proliferator-activated receptor gamma coactivator 1 alpha [28]. Consequently, serum levels of ZAG have been reported to be significantly decreased in subjects with obesity, showing negative correlations with anthropometric measures such as BMI, waist circumference, waist-to-hip ratio, body fat percentage, and triglycerides, while positively correlating with HDL cholesterol and adiponectin [3,28,29,30]. Additionally, a significant reduction in ZAG expression in the subcutaneous white adipose tissue of individuals with obesity has been found in both Caucasian and Chinese populations [3,28,31,32,33]. These results appeared to be independent of the habitual dietary fat intake and physical activity of the subjects included [33]. However, despite a 9.6% body weight loss after 6 months in 13 subjects, they failed to demonstrate significant changes in ZAG mRNA levels in abdominal subcutaneous adipose tissue [34].

ZAG is an adipokine involved in the regulation of lipid and glucose metabolism, as well as the control of fat mass and energy expenditure [35]. Our results suggest that, in individuals without obesity or type 2 diabetes, postprandial insulin plays a key role in inhibiting ZAG’s lipolytic actions, thereby facilitating adipogenesis. This finding underscores how, after a liquid nutritional supplement, insulin not only helps regulate blood sugar levels but also promotes fat storage by suppressing ZAG’s ability to break it down. In essence, this may be considered an additional mechanism through which insulin exerts its anabolic effect, shifting the body’s metabolism toward energy storage rather than energy expenditure.

To the best of our knowledge, the rapid downregulation of ZAG levels by insulin has not been previously described. The molecular mechanism through which insulin quickly reduces ZAG expression warrants further research, specifically with broader cohorts with metabolic diseases, mechanistic studies, and potential therapeutic targets. Insulin acts on target cells through its specific insulin receptor (IR), which consists of two α subunits located outside the cell membrane and two β subunits anchored in the cell membrane. Activation of the IR causes the autophosphorylation of the β subunits, thereby initiating two major signal transduction pathways: the PI3K-PKB/Akt-dependent pathway and the MAPK (mitogen-activated protein kinase) pathway [36]. Therefore, the fast downregulation of ZAG induced by insulin may occur through various transcription factors associated with insulin signaling cascades, such as Akt, mTOR, FOXO1, or ERK-1/2 [37]. In addition, the mechanisms through which insulin downregulates ZAG expression may also involve its ability to modulate several cytokines, as demonstrated by findings that insulin administration enhances both mRNA expression and protein secretion of TNF-alpha in a dose-dependent manner in macrophages [38]. ZAG, in turn, is recognized as an anti-inflammatory adipokine that protects against inflammation, showing a strong negative correlation with insulin resistance and leptin secretion while stimulating the production of the anti-inflammatory adipokine adiponectin [4,33]. Although we did not specifically analyze these transcription factors in this study, future work may explore their role in modulating ZAG expression. While the early reduction in ZAG mRNA at 15 min may seem unexpected, similar rapid transcriptional responses to insulin have been observed in hepatocytes and HepG2 cells [39,40,41]. This supports the plausibility of a primary, insulin-mediated transcriptional mechanism.

Therefore, further research in needed to better characterize the ZAG promoter region and identify the transcription factors involved in insulin regulation. It is worth mentioning that ZAG has been reported to inhibit insulin-induced glucose uptake in human adipocytes by impairing insulin signaling at the AKT level in a β2 adrenergic receptor- and protein phosphatase 2A-dependent manner [42].

## 4. Materials and Methods

### 4.1. Subjects

A total of 24 apparently healthy individuals, who were partners of consecutive patients attending the outpatient obesity unit at our university hospital, were included in this study. Participants were eligible if they were over 18 years of age and did not have obesity, defined as a BMI of less than 30 kg/m^2^, and who did not receive medication or other interventions. Exclusion criteria included the presence of metabolic conditions such as type 2 diabetes [defined by fasting blood glucose levels of 126 mg/dL or higher, or 200 mg/dL or higher after an oral OGTT], chronic kidney disease, chronic intestinal diseases or history of bariatric surgery, chronic liver disease, active neoplasia, or active alcohol consumption exceeding 40 g/day for men and 20 g/day for women. All participants provided written informed consent, and the study protocol was approved by the hospital’s human ethics committee [PR(AG)690/2020].

### 4.2. Laboratory Assessments

Following a 12 h overnight fast, the response of ZAG to three insulin stimulatory dynamic tests (ISDTs) was evaluated over a 14-day period. These tests included (1) an OGTT with 75 g of glucose, (2) a standardized liquid nutritional supplement test (200 mL, 150 kcal; containing 6 g of protein, 5.8 g of fat, and 18.4 g of carbohydrates; Fortisip; Nutricia, Madrid, Spain) administered over a 10 min period, and (3) the intravenous administration of 1 mg of glucagon (Novo Nordisk Pharma, Bagsvaerd, Denmark). The liquid nutritional supplement was selected instead of a regular solid meal due to its standardized macronutrient composition, ease of administration, and reproducibility across participants, ensuring consistent experimental conditions across all subjects.

Blood samples were collected from the antecubital vein at specific time intervals tailored to each test: at 0, 30, 60, 90, and 120 min for the OGTT; at 0, 30, 60, 90, 120, and 180 min for the standardized liquid nutritional supplement test; and at 0, 5, and 15 min for the glucagon test. In each of these samples, levels of ZAG, insulin, C-peptide, and glucose were measured. Serum insulin levels were measured using a radioimmunoassay (INSI-CTK IRMA; DiaSorin, Reutlinger, Germany), while serum C-peptide levels were determined through a competitive immunoassay (C-PEP-RIA-CT; BioSource Europe, Nivelles, Belgium).

### 4.3. HepG2 Cultures

HepG2 hepatoblastoma cells, generously provided by Dr. G. Hammond (University of British Columbia, Vancouver, BC, Canada), were routinely cultured in low-glucose DMEM (catalog No. 11885-084, Thermo Fisher Scientific, Waltham, MA, USA) supplemented with 10% fetal bovine serum and antibiotics (100 U/mL penicillin and 100 µg/mL streptomycin). For the experiments, HepG2 cells were grown to 80% confluence in low-glucose DMEM and then treated with either insulin (20 µIU/mL) for varying durations over a 2 h period (0, 15, 30, 60, and 120 min). At the conclusion of the experiment, the media were collected for analysis, and the cells were harvested for RNA and protein extraction.

HepG2 cells were chosen as an established model of human hepatocytes. Although ZAG is predominantly expressed in adipose tissue, previous work from our group demonstrated that ZAG is also expressed in the human liver and in HepG2 cells, making this a suitable model for studying acute hepatic regulation of ZAG [3].

### 4.4. Total RNA Preparation and Real-Time PCR

After treatment, total RNA was extracted from HepG2 cells using TRIzol reagent (Invitrogen, Barcelona, Spain). RNA concentrations were determined by measuring absorbance at 260 nm (A260), and purity was assessed by calculating the A260/A280 ratio. Reverse transcription was performed at 42 °C for 50 min using 3 µg of total RNA and 200 U of Superscript II, in combination with an oligo-dT primer and reagents supplied by Invitrogen. An aliquot of the resulting cDNA was then amplified in a 25 µL reaction using Power SYBR Green PCR master mix (Invitrogen) and specific oligonucleotide primer pairs targeting human ZAG (forward primer: 5′-CTTGGCTCACTCAATGACCTC, reverse primer: 5′-CTCCGCTGCTTCTGTTATTC) and human 18S rRNA (forward primer, 5′-TAACGAACG AGACTCTGGCAT, reverse primer: 5′-CGGACATCTAAGGGCATCACAG) as previously described by Selva et al. [3]. The PCR amplifications were performed according to the manufacturer’s recommendations (Applied Biosystems, Foster City, CA, USA), including an initial denaturation at 95 °C for 10 min, followed by 40 cycles of denaturation at 95 °C for 15 s and annealing/extension at 60 °C for 1 min.

### 4.5. Western Blot Analysis

After treatment, proteins were extracted from HepG2 cells using a lysis buffer consisting of 50 mM Tris-HCl (pH 7.5), 150 mM NaCl, 2 mM EDTA, 1% Nonidet P-40, and 0.1% sodium dodecyl sulfate (SDS), supplemented with complete protease inhibitor cocktail (Roche Diagnostics, Barcelona, Spain). The extraction process was carried out at 4 °C, followed by centrifugation at 12,000 rpm (13,500× *g*) at 4 °C for 10 min to obtain the total protein extracts.

Protein extracts were subjected to Western blot analysis using antibodies against human ZAG (1E2; catalogue No. sc-21720; Santa Cruz Biotechnology Inc., Santa Cruz, CA, USA; dilution 1:1000) and human cyclophilin A (SA-296; BIOMOL Inc., Plymouth Meeting, PA, USA; dilution 1:10,000). Specific antibody–antigen complexes were detected using horseradish-peroxidase-labeled secondary antibodies, either rabbit anti-mouse IgG or goat anti-rabbit IgG, followed by chemiluminescent substrate development (Pierce Biotechnology Inc., Rockford, IL, USA). The resulting signals were visualized by exposure to X-ray film.

### 4.6. ELISA Analysis

ZAG levels were measured using an ELISA kit (BioVendor, Heidelberg, Germany). The assay had a lower detection limit of 0.3 µg/mL. The intra-assay and inter-assay coefficients of variation were 4.7 and 6.6%, respectively.

### 4.7. Statistical Analysis

The normal distribution of the variables was evaluated using the Kolmogorov–Smirnov test. Comparison of quantitative variables was performed by either Student’s *t* test or the Mann–Whitney test according to the data distribution. All data are presented as mean ± standard deviation. Significance was accepted at the level of *p* < 0.05. Statistical analyses were performed with the SPSS statistical package (IBM SPSS Statistics for Windows, Version 20.0, IBM Corp., Armonk, NY, USA).

## 5. Conclusions

In conclusion, our results highlight the delicate balance between insulin signaling and fat metabolism in maintaining the overall energy balance. Increased insulin concentrations, both in a feeding state and in cultured HepG2 cells, promote the acute downregulation of ZAG expression and production, reinforcing the anabolic role of insulin. These findings support the role of insulin in the acute regulation of ZAG and highlight the need for further mechanistic studies, including both adipocyte- and liver-based models, to dissect tissue-specific regulatory pathways.

## Figures and Tables

**Figure 1 ijms-26-05438-f001:**
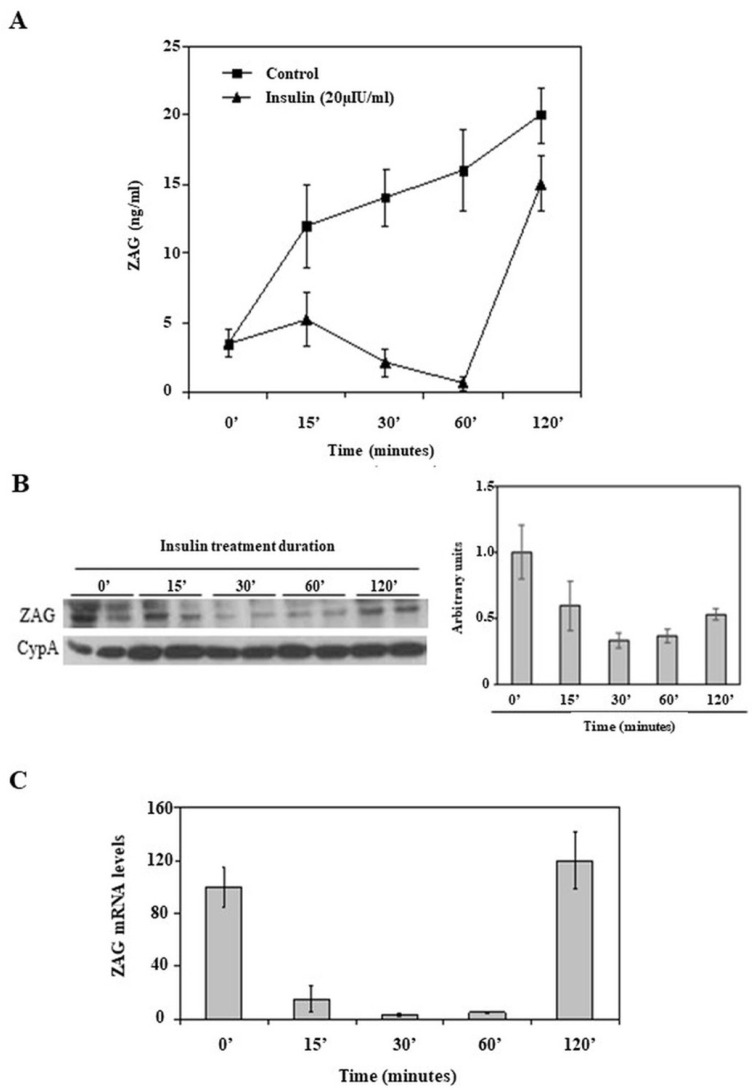
Acute insulin downregulation of ZAG production by HepG2 cells. HepG2 cells were treated in a 2 h experiment with vehicle or insulin (20 µIU/mL) at different time points. (**A**) ZAG media from HepG2 cells were measured over the course of a 2 h experiment, treated with vehicle or insulin (20 µIU/mL), measured by ELISA. Data are expressed as mean ± SD of triplicates. (**B**) Representative Western blot images of ZAG and CypA from extracts of HepG2 cells treated with insulin, as described in (**A**). No untreated control condition was included in this experiment. (**C**) ZAG mRNA levels from HepG2 cells treated with insulin, as in (**A**). Human 18S was amplified as internal control, and values are expressed as percentage relative to untreated cells. Data are expressed as mean ± SD of triplicates.

**Table 1 ijms-26-05438-t001:** Clinical features and biochemical parameters of the subjects included in this study.

n	24
Age (years)	53.0 ± 16.9
Sex (male/female)	12/12
BMI (kg/m^2^)	24.8 ± 4.9
Fasting glucose (mg/dL)	100.4 ± 10.8
Insulin (pmol/L)	11.7 ± 5.5
C-peptide (pmol/L)	4.2 ± 6.1

Data are mean ± SD.

**Table 2 ijms-26-05438-t002:** ZAG, insulin, C-peptide, and glucose values during the oral glucose tolerance test (OGTT) with 75 g of glucose, the intake of a standardized liquid nutritional supplement, and the intravenous administration of 1 mg of glucagon.

		**0 min**	**30 min**	**60 min**	**90 min**	**120 min**
OGTT	ZAG (µg/mL)	42.1 ± 18.3	38.3 ± 15.8	36.5 ± 20.2	35.2 ± 19.7 ^a^	40.4 ± 21.9
	Insulin (pmol/L)	13.3 ± 7.7	73.7 ± 43.2 ^b^	84.3 ± 50.5 ^b^	118.3 ± 80.7 ^b^	104.2 ± 70.3 ^b^
	C-peptide (pmol/L)	2.9 ± 0.8	6.5 ± 2.7 ^b^	9.8 ± 4.6 ^b^	12.2 ± 4.6 ^b^	12.5 ± 4.3 ^b^
	Glucose (mmol/L)	94.1 ± 7.6	167.5 ± 31.0 ^b^	187.5 ± 44.9 ^b^	176.0 ± 53.2 ^b^	150.4 ± 46.0 ^b^
		**0 min**	**60 min**	**90 min**	**120 min**	**180 min**
Standardized liquid nutritional supplement	ZAG (µg/mL)	42.3 ± 20.3	38.1 ± 18.8	37.4 ± 16.5	31.5 ± 14.9 ^a^	38.8 ± 25.6
	Insulin (pmol/L)	13.3 ± 8.1	68.2 ± 39.6 ^b^	51.7 ± 36.1 ^b^	44.1 ± 37.1 ^b^	18.3 ± 10.8 ^a^
	C-peptide (pmol/L)	3.3 ± 1.2	8.6 ± 3.6 ^b^	7.5 ± 2.8 ^b^	7.0 ± 2.9 ^b^	4.7 ± 1.6 ^b^
	Glucose (mmol/L)	102.4 ± 10.2	129.4 ± 27.4 ^b^	115.2 ± 28.5 ^a^	104.3 ± 23.8	90.0 ± 10.8 ^b^
		**0 min**	**5 min**	**15 min**		
1 mg Glucagon	ZAG (µg/mL)	42.2 ± 26.7	34.4 ± 17.0 ^a^	38.2 ± 20.1		
	Insulin (pmol/L)	11.7 ± 5.5	74.4 ± 34.8 ^b^	50.0 ± 23.4 ^b^		
	C-peptide (pmol/L)	4.2 ± 6.1	6.7 ± 3.2	5.8 ± 2.6		
	Glucose (mmol/L)	100.4 ± 10.8	113.8 ± 10.0 ^b^	131.9 ± 16.3 ^b^		

Data are mean ± SD. ^a^: *p* < 0.05; ^b^: *p* < 0.001. *p*-values indicate comparisons between each value and its corresponding baseline.

## Data Availability

The datasets generated for this study are available upon request from the corresponding author.

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
