# Peer review of "Acute Downregulation of Zinc α2-Glycoprotein: Evidence from Human and HepG2 Cell Studies"

_ijms, 2025, doi:10.3390/ijms26125438_

Round 1
Reviewer 1 Report (New Reviewer)
Comments and Suggestions for Authors
Lecube et al. studied the relationship of ZAG and insulin levels in human, after three different overload. The authors complemented those studies with in vitro experiments using HepG2 cells line.
The introduction include an overall review of function of ZAP already described, however it do not include regulation of hepatocyte lipid metabolism, issue included in their research.
The results and methods are appropriated described.
The discussion could be improved, centering in the production fo ZAG by hepatocyte, its effect in liver carbohydrates/lipids metabolism and relation with systemic changes, its relevance. The introduction gives evidence of the ZAP central role in adipocyte metabolism, but the article include assays of insulin treatment in HepG2 cells. Could also include assay with glucagon. Although there is relevant know the response of ZAP gene to insulin, the proposal of transcriptional response to insulin in HepG2, could not be plausible, because it is observed even at 15 minutes. The conclusion and projections are appropriates
Minor correction:
-in line 96, check the sentence "ZAG levels showed a progressive decrease in ZAG"
- The original image does not include a full size imagen of wester-blot
Author Response
We would like to thank the reviewer for their thoughtful and constructive comments about our manuscript “Acute downregulation of Zinc α2-glycoprotein: Evidence from Human and HepG2 Cell Studies”. We have carefully considered each suggestion and have revised the manuscript accordingly. Below we provide our detailed responses.
Comment1 : The introduction include an overall review of function of ZAP already described, however it do not include regulation of hepatocyte lipid metabolism, issue included in their research.
Thank you for your helpful comment. We agree with your observation and have added a paragraph at the end of the Introduction to specifically highlight the potential role of ZAG in the regulation of hepatic lipid metabolism and the rationale for studying its modulation in hepatocytes.
Comment 2: The results and methods are appropriated described.
Thank you for your positive feedback. We appreciate your recognition of the clarity and adequacy of our methods and results.
Comment 3: The discussion could be improved, centering in the production of ZAG, its effect on liver carbohydrate/lipids metabolism and relation with systemic changes.
Thank you. We have revised the Discussion to better highlight the role of hepatic ZAG in carbohydrate and lipid metabolism, and its systemic implications. We now discuss how insulin’s inhibitory effect on hepatic ZAG may contribute to energy storage following food intake.
Comment 4: The introduction gives evidence of the ZAP central role in adipocyte metabolism, but the article include assays of insulin treatment in HepG2 cells. Could also include assay with glucagon?
We appreciate the reviewer’s suggestion. In this study, glucagon was administered in vivo as part of a dynamic test designed to stimulate endogenous insulin secretion. Our primary aim was to assess the acute modulation of circulating ZAG levels in response to metabolic challenges rather than to isolate the individual effects of each hormone in vitro. While in vitro assays with glucagon could provide additional insights, they fall beyond the scope of the current study. Nonetheless, we have clarified in the revised manuscript that the observed reduction in ZAG following glucagon administration is likely mediated by the associated insulin response, rather than a direct effect of glucagon itself.
Comment 5: Although there is relevant know the response of ZAP gene to insulin, the proposal of transcriptional response to insulin in HepG2, could not be plausible, because it is observed even at 15 minutes.
We acknowledge the reviewer’s concern. However, rapid transcriptional responses following insulin stimulation have been documented in hepatic models, including HepG2 cells. For example, changes in the expression of certain insulin-responsive genes have been reported within 15–30 minutes of exposure (Saltiel AR et al., 2001; Patti ME et al. 2003; Gerhart-Hines Z et al., 2011). In our study, the early reduction in ZAG mRNA levels may reflect immediate-early gene regulation or rapid mRNA destabilization mechanisms. We have addressed this point in the Discussion section of the revised manuscript and supporting references have been added.
Comment 6: The conclusion and projections are appropriates.
We appreciate your positive evaluation.
Comment 7: Minor correction: in line 96, check the sentence "ZAG levels showed a progressive decrease in ZAG"
Thank you for pointing out this redundancy. We have revised the sentence for clarity.
Comment 8: The original image does not include a full size imagen of wester-blot
Unfortunately, we did not find the original film of the western blot used in the figure. However, we have found the film from another experiment performed in the same conditions. We have modified the figure accordingly with the new image. We also scanned the films from where the images were cropped. As you will see, in the original film, there are four lanes more of longer time treatments, but we decided not to show them.
Reviewer 2 Report (New Reviewer)
Comments and Suggestions for Authors
The authors wrote a manuscript where they show that insulin levels in humans inversely correlate with the levels of Zinc-alpha2-glycoprotein (ZAG), a soluble glycoprotein that has key roles in lipid metabolism. In particular, the authors investigate the acute effects of insulin on ZAG levels both in vivo and in vitro.
The manuscript is well written and the topic is of interest for researchers in the field, especially because previous studies have shown contradictory results in humans. However, in the current version, the manuscript does not offer a comprehensive analysis, and additional analysis would be required to improve its quality.
These suggestions below may help the authors improving the quality of the manuscript:
- The authors have previously shown that no relationship was found between ZAG and homeostasis model assessment for insulin resistance and insulin had no effect on ZAG production in vitro using the same cell line (HepG2) (PMID: 19622624); however, in their work they only tested chronic insulin administration (5 consecutive days), while in this work they evaluate the effects of insulin after acute dosing. Could they speculate in more details why chronic administration did not show effects in their previous studies (e.g. which molecular pathways could be involved in the insulin-mediated ZAG suppression in acute vs chronic exposure)?
- The authors state in their abstract that their results suggest that insulin acutely inhibits ZAG production, but in their results they only show a correlation between high insulin and low ZAG levels. Since there could be other hormones involved in ZAG downregulation following ISDTs, the authors should either show a convincing direct link/molecular pathway connecting insulin with ZAG downregulation or be more cautious in their statements.
- Western blot data in Figure 1b is not represented with high quality – could the authors please show a picture of their results where the signal is not saturated, and protein bands are more clearly shown?
- The authors state in their introduction that ZAG is primarily produced in mature adipocytes, where it exerts its lipolytic and browning effects; however, they choose HepG2 for their in vitro assessments, which is a hepatocellular carcinoma cell line. Could they please justify why this cell line was specifically chosen, instead of adipocytes?
- Following up on my point above, the authors should assess the effects of insulin in an in vitro model using mature adipocytes (e.g. 3T3-L1 differentiated cells, or similar models); it would be relevant to assess whether the insulin effects shown in figure 1 are recapitulated in adipocytes.
- The authors speculate about the possible molecular mechanisms involved in the insulin-mediated effects on ZAG protein modulation, and mention a few possible mediators (lines 221-224); however, it would be relevant to analyse these mechanisms in details. Could the authors test the expression levels of PI3K/mTOR/MAPK downstream TFs (e.g. FOXOs), and verify whether these are directly linking insulin signalling with ZAG downregulation?
- Hyperglycaemia is known to activate the inflammatory cytokine production, and TNF-alpha has been shown to negatively regulate ZAG levels. Can the authors verify whether the ZAG modulation they observe is not due to increased levels of TNF-alpha, instead of hyperinsulinemia?
Author Response
We would like to thank the reviewer for their thoughtful and constructive comments about our manuscript “Acute downregulation of Zinc α2-glycoprotein: Evidence from Human and HepG2 Cell Studies”. We have carefully considered each suggestion and have revised the manuscript accordingly. Below we provide our detailed responses.
Comment 1: The authors have previously shown that no relationship was found between ZAG and homeostasis model assessment for insulin resistance and insulin had no effect on ZAG production in vitro using the same cell line (HepG2) (PMID: 19622624); however, in their work they only tested chronic insulin administration (5 consecutive days), while in this work they evaluate the effects of insulin after acute dosing. Could they speculate in more details why chronic administration did not show effects in their previous studies (e.g. which molecular pathways could be involved in the insulin-mediated ZAG suppression in acute vs chronic exposure)?
Thank you for pointing this out. In our previous study (Selva et al., JCEM 2009), HepG2 cells were exposed to insulin for five consecutive days, and ZAG expression was assessed 24 hours after the last treatment. This long exposure likely activated compensatory mechanisms or feedback loops that attenuate insulin signaling, masking any immediate effect. In contrast, the current study focuses on acute effects within a short window (15 to 120 minutes), where primary signaling events such as PI3K/Akt or MAPK activation might have a direct influence on ZAG expression before secondary adaptations occur. We have clarified this difference in the revised Discussion.
Comment 2: The authors state in their abstract that their results suggest that insulin acutely inhibits ZAG production, but in their results they only show a correlation between high insulin and low ZAG levels. Since there could be other hormones involved in ZAG downregulation following ISDTs, the authors should either show a convincing direct link/molecular pathway connecting insulin with ZAG downregulation or be more cautious in their statements.
We agree with the reviewer that our in vivo results are showing just correlation. However, our in vitro data clearly show a rapid and transient downregulation of ZAG mRNA and protein levels after insulin treatment in HepG2 cells, supporting a direct effect. We have tempered the wording in the Abstract and Discussion to emphasize that the in vivo data are associative, while the in vitro data suggest a direct regulatory effect.
Comment 3: Western blot data in Figure 1b is not represented with high quality – could the authors please show a picture of their results where the signal is not saturated, and protein bands are more clearly shown?
Unfortunately, we did not find the original film of the western blot used in the figure. However, we did find the film from another experiment performed in the same conditions. We have modified the figure accordingly with the new image. We also have scanned the films from where the images were cropped. As you will see, in the original film, there are for lanes more of longer time treatments but we decided not to show them.
Comment 4: The authors state in their introduction that ZAG is primarily produced in mature adipocytes, where it exerts its lipolytic and browning effects; however, they choose HepG2 for their in vitro assessments, which is a hepatocellular carcinoma cell line. Could they please justify why this cell line was specifically chosen, instead of adipocytes?
This is a very interesting point. We are interested in the role of ZAG in the pathophysiology of MALD and for this reason we are examining mechanistic regulation using HepG2. We realize that HepG2 cells do not fully replicate the physiology of primary hepatocytes, but they provide a consistent and reliable system to study acute transcriptional responses. Given the rapid effects observed in our current study, this model was well suited for investigating the early response of ZAG to insulin. We now clarify this rationale in the revised version of the manuscript.
Comment 5: Following up on my point above, the authors should assess the effects of insulin in an in vitro model using mature adipocytes (e.g. 3T3-L1 differentiated cells, or similar models); it would be relevant to assess whether the insulin effects shown in figure 1 are recapitulated in adipocytes.
We fully agree that investigating insulin’s effects on ZAG in adipocytes would be a valuable addition. However, the current study was specifically designed to explore acute regulation of ZAG in humans and in a hepatic model. Our group has previously studied ZAG expression in human adipose tissue and demonstrated its downregulation in obesity, independently of insulin resistance (Selva et al., JCEM 2009). The present work complements those findings by focusing on the dynamic regulation of ZAG in response to acute metabolic stimuli. We consider these results an important step toward a more comprehensive understanding of ZAG biology and believe they provide a solid basis for future studies using adipocyte-based models.
Comment 6: The authors speculate about the possible molecular mechanisms involved in the insulin-mediated effects on ZAG protein modulation, and mention a few possible mediators (lines 221-224); however, it would be relevant to analyse these mechanisms in details. Could the authors test the expression levels of PI3K/mTOR/MAPK downstream TFs (e.g. FOXOs), and verify whether these are directly linking insulin signalling with ZAG downregulation?
We appreciate the reviewer’s suggestion and agree this would help elucidate the underlying mechanisms. Unfortunately, these analyses fall outside the scope of the current study. However, we now discuss these pathways more thoroughly in the revised Discussion, emphasizing that transcription factors such as FOXO1, ERK1/2 and mTOR may mediate insulin’s effects on ZAG.
Comment 7: Hyperglycaemia is known to activate the inflammatory cytokine production, and TNF-alpha has been shown to negatively regulate ZAG levels. Can the authors verify whether the ZAG modulation they observe is not due to increased levels of TNF-alpha, instead of hyperinsulinemia?
This is an important consideration. While we did not measure TNF-alpha levels in this study, all in vivo assessments were conducted in healthy individuals without obesity or inflammatory conditions. Thus, it is unlikely that circulating TNF-alpha significantly contributed to the observed acute ZAG modulation. Furthermore, in our previous study (Selva et al., JCEM 2009) we showed that TNF-alpha downregulates ZAG in HepG2 cells, but insulin alone had no chronic effect, highlighting that different regulatory mechanisms are likely involved in acute versus chronic settings.
Round 2
Reviewer 2 Report (New Reviewer)
Comments and Suggestions for Authors
The authors have replied to the reviewer's comments; however, a couple of points should be addressed before manuscript acceptance:
- Related to answer in comment 4: if the authors are interested in the pathophysiology of MAFLD, they should adjust the abstract accordingly (otherwise as it is currently written it seems they aim to analyze ZAG's role in lipid metabolism into adipocytes, where the protein is primarily produced)
- Related to comment 7: hyperglycemia activates inflammatory cytokine production regardless of the nature of the individuals (even if individuals were not obese); therefore, I still think TNF-alpha measurements in vivo would be an important addition to the manuscript. The authors should clarify whether they have enough samples stored from this study to measure it; if they don't, they should at least address/discuss this issue in the manuscript.
Author Response
ANSWERS TO REVIEWER 2
We would like to thank again the reviewer for their thoughtful and constructive comments about our manuscript “Acute downregulation of Zinc α2-glycoprotein: Evidence from Human and HepG2 Cell Studies”. We have carefully considered each suggestion and have revised the manuscript accordingly. Below we provide our detailed responses.
[Comment 1] Related to answer in comment 4: if the authors are interested in the pathophysiology of MAFLD, they should adjust the abstract accordingly (otherwise as it is currently written it seems they aim to analyze ZAG's role in lipid metabolism into adipocytes, where the protein is primarily produced)
We thank the reviewer for this helpful suggestion. In response, we have made three minor but meaningful adjustments to clarify the potential relevance of our findings to the pathophysiology of MAFLD. In the Abstract, we now mention that ZAG is produced not only in adipocytes but also in hepatocytes, and we highlight the potential implications of our results for metabolic conditions such as MAFLD. We have also clarified in the Introduction that ZAG is expressed in various human tissues, including the liver. Finally, in the Conclusion, we emphasize the importance of future mechanistic studies using both adipocyte and liver-based models to explore tissue-specific regulation. These changes better reflect the scope of our work without altering its main focus.
[Comment 2]: Related to comment 7: hyperglycemia activates inflammatory cytokine production regardless of the nature of the individuals (even if individuals were not obese); therefore, I still think TNF-alpha measurements in vivo would be an important addition to the manuscript. The authors should clarify whether they have enough samples stored from this study to measure it; if they don't, they should at least address/discuss this issue in the manuscript.
We thank the reviewer for the additional observation regarding the potential involvement of inflammatory cytokines, such as TNF-α, in the acute metabolic response to hyperglycemia. In the revised manuscript, we have expanded the discussion to acknowledge insulin’s known ability to stimulate TNF-α expression, and we highlight the anti-inflammatory role of ZAG and its interplay with adipokines such as adiponectin and leptin. These additions help to contextualize the relationship between insulin, inflammation, and ZAG regulation. While we agree that in vivo TNF-α measurements would provide valuable mechanistic insights, unfortunately, no additional serum samples are available from this study to perform such analyses retrospectively.
This manuscript is a resubmission of an earlier submission. The following is a list of the peer review reports and author responses from that submission.
Round 1
Reviewer 1 Report
Comments and Suggestions for Authors
This study provides novel insights into the acute regulation of ZAG by insulin and its potential role in energy balance. While the findings are significant, expanding the research with broader cohorts and mechanistic studies would strengthen its impact. The manuscript is well-structured, but minor improvements in methodology and figure clarity would enhance its robustness.
The references are appropriate and cite relevant studies on ZAG, insulin signaling, and metabolic regulation. However, some recent studies on insulin’s role in adipokine regulation could further support the discussion.
Tables: The data presentation is clear, but some p-values are not fully annotated for clarity.
Figures: The Western blot images and ELISA data are informative. However, adding representative blots for all experimental conditions in the figure legends would enhance transparency.
Further investigation into the transcription factors and signaling pathways mediating ZAG downregulation by insulin is needed. Including individuals with insulin resistance or type 2 diabetes would clarify the clinical significance of the findings.
Summary:The topic is highly relevant to metabolic research, as ZAG is an adipokine involved in lipid metabolism and energy homeostasis. The study addresses a knowledge gap by examining the acute regulatory effects of insulin on ZAG, a topic that has been explored primarily in animal models but remains underexplored in humans. This makes the research original and valuable in the context of metabolic disorders, insulin resistance, and potential therapeutic targets.
Comments on the Quality of English Language
- The language is formal and appropriate for a scientific article.
- The key ideas are clearly conveyed, and the structure is logical.
- Technical terms and scientific concepts are well-explained.
- Some sentences are overly complex and could be restructured for clarity.
- Example: "In vivo results obtained in our study confirm that insulin acutely downregulates ZAG expression and production, as evidenced by the similar pattern observed following insulin administration in HepG2 cells."
- Suggestion: "Our in vivo results confirm that insulin acutely downregulates ZAG expression and production. This is supported by a similar pattern observed in HepG2 cells following insulin administration."
Author Response
Thank you very much for the revision process of our manuscript entitled “Acute downregulation of Zinc α2-glycoprotein by insulin in subjects without obesity” (Ms # ijms-3480660).
1.- This study provides novel insights into the acute regulation of ZAG by insulin and its potential role in energy balance. While the findings are significant, expanding the research with broader cohorts and mechanistic studies would strengthen its impact. The manuscript is well-structured, but minor improvements in methodology and figure clarity would enhance its robustness.
We have acknowledged the need for further research, specifically involving broader cohorts and mechanistic studies, and have included this point in the final paragraph of the Discussion section.
2.- The references are appropriate and cite relevant studies on ZAG, insulin signaling, and metabolic regulation. However, some recent studies on insulin’s role in adipokine regulation could further support the discussion.
Following the reviewer’s suggestion, we have added a recent review by Gilani et al. in the first paragraph of the Discussion section. This study highlights how insulin-mediated adipose tissue signaling influences distal organ health, reinforcing the interconnected nature of metabolic pathways.
Gilani A, Stoll L, Homan EA, Lo JC. Adipose Signals Regulating Distal Organ Health and Disease. Diabetes. 2024;73(2):169-177. doi: 10.2337/dbi23-0005.
3.- Tables: The data presentation is clear, but some p-values are not fully annotated for clarity.
We have clarified in the table legend that p-values indicate comparisons between each value and its corresponding baseline.
4.- Figures: The Western blot images and ELISA data are informative. However, adding representative blots for all experimental conditions in the figure legends would enhance transparency.
We appreciate the reviewer’s suggestion regarding the inclusion of representative blots for all experimental conditions. As this study exclusively evaluated the effect of insulin treatment on HepG2 cells, without including an untreated control group, we have clarified this in the figure legend to ensure transparency.
5.- Further investigation into the transcription factors and signaling pathways mediating ZAG downregulation by insulin is needed. Including individuals with insulin resistance or type 2 diabetes would clarify the clinical significance of the findings.
We agree with the reviewer that our initial results warrant further investigation into the transcription factors and signaling pathways mediating ZAG downregulation by insulin. Future studies could include individuals with insulin resistance or type 2 diabetes to clarify the clinical relevance of our findings. We have added this reflection in the last paragraph of the Discussion section.
6.- Summary: The topic is highly relevant to metabolic research, as ZAG is an adipokine involved in lipid metabolism and energy homeostasis. The study addresses a knowledge gap by examining the acute regulatory effects of insulin on ZAG, a topic that has been explored primarily in animal models but remains underexplored in humans. This makes the research original and valuable in the context of metabolic disorders, insulin resistance, and potential therapeutic targets.
We thank the reviewer for this positive comment.
7.- Comments on the Quality of English Language: The language is formal and appropriate for a scientific article / The key ideas are clearly conveyed, and the structure is logical / Technical terms and scientific concepts are well-explained. Some sentences are overly complex and could be restructured for clarity. Example: "In vivo results obtained in our study confirm that insulin acutely downregulates ZAG expression and production, as evidenced by the similar pattern observed following insulin administration in HepG2 cells." Suggestion: "Our in vivo results confirm that insulin acutely downregulates ZAG expression and production. This is supported by a similar pattern observed in HepG2 cells following insulin administration."
We have revised the text accordingly and simplified complex sentences for improved clarity.
Reviewer 2 Report
Comments and Suggestions for Authors
Major concern:
Zinc alpha 2-glycoprotein expression is regulated by glucocorticoids. Due to its high sequence homology with lipid-mobilizing factor and high expression in cancer cachexia, it is considered as a novel adipokine. Recently it has been reported that Zinc alpha 2-glycoprotein is associated with insulin resistance in humans and is regulated by hyperglycemia and hyperinsulinemia. Circulating ZAG correlated positively with HDL cholesterol and adiponectin, and correlated inversely with BMI, waist-to-hip ratio, body fat percentage, triglycerides, fasting blood glucose, fasting insulin, HbA1c, and homeostasis model assessment of insulin resistance (HOMA-IR).
The aim of this study was to investigate the response of circulating ZAG to three acute stimuli (oral glucose overload, a standard meal, and intravenous glucagon) in individuals without obesity and diabetes. However, the contribution is limited.
This result of this study can be predictable and this study does not reveal novel findings. In addition, the conclusion is weak.
Author Response
Thank you very much for the revision process of our manuscript entitled “Acute downregulation of Zinc α2-glycoprotein by insulin in subjects without obesity” (Ms # ijms-3480660).
1.- Zinc alpha 2-glycoprotein expression is regulated by glucocorticoids. Due to its high sequence homology with lipid-mobilizing factor and high expression in cancer cachexia, it is considered as a novel adipokine. Recently it has been reported that Zinc alpha 2-glycoprotein is associated with insulin resistance in humans and is regulated by hyperglycemia and hyperinsulinemia. Circulating ZAG correlated positively with HDL cholesterol and adiponectin, and correlated inversely with BMI, waist-to-hip ratio, body fat percentage, triglycerides, fasting blood glucose, fasting insulin, HbA1c, and homeostasis model assessment of insulin resistance (HOMA-IR).
The aim of this study was to investigate the response of circulating ZAG to three acute stimuli (oral glucose overload, a standard meal, and intravenous glucagon) in individuals without obesity and diabetes. However, the contribution is limited.
This result of this study can be predictable and this study does not reveal novel findings. In addition, the conclusion is weak.
We thank the reviewer for their insightful comments and reflections on our study. We acknowledge the complex regulatory mechanisms of Zinc alpha 2-glycoprotein (ZAG), including its association with insulin resistance and its correlations with various metabolic parameters, as highlighted in recent literature. While our study focuses on individuals without obesity or diabetes, we believe that our findings contribute to the understanding of the acute effects of insulin on ZAG regulation in this specific population.
We agree that further research, including studies involving individuals with insulin resistance or metabolic disorders, will be essential to fully elucidate the role of ZAG in metabolic homeostasis. We are committed to deepening this line of investigation in future studies to strengthen the clinical relevance and mechanistic insights of our work. We appreciate the reviewer’s valuable input, which will help guide our future research directions.
Reviewer 3 Report
Comments and Suggestions for Authors
Manuscript titled “Acute downregulation of Zinc α2-glycoprotein by insulin in subjects without obesity” reports in vitro and human-derived data regarding the effects of insulin and other stimuli on zinc α2-glycoprotein (ZAG). The data is interesting, although some aspects of the manuscript could be improved:
- The manuscript’s title suggests that the observed effects on ZAG were due to insulin in human subjects. This is slightly misleading since, strictly speaking, you administered insulin to HepG2 cells, but not to the participants. They were instead administered a glucose load, a macronutrient load or glucagon. Although human and in vitro data agree with each other, the stimuli are not the same on both models. Thus, please consider rephrasing the title to more adequately reflect your methods.
- Lines 65-70 mention that ZAG has been shown to decrease in obese individuals, and “suggesting a negative correlation with insulin resistance” (line 69). As currently written, the information suggests that obesity negatively correlates with ZAG, while insulin resistance is not specifically mentioned. Thus, please consider specifically mentioning that insulin resistance is present in the individuals of the cited papers, or state that ZAG negatively correlates with obesity, whichever is the most appropriate.
- The abstract and Section 4 “Laboratory assessments” (and elsewhere in the manuscript) mention a “standard meal” or “standard food”, however, the “Fortisip” product administered appears to be more of a ready-to-drink product instead of a solid meal per se. Please consider providing a better description and/or nomenclature of the product in order to make its nature clear to the reader. Upon first reading your manuscript, I thought the meal used was an actual solid meal, and not a liquid product, which is why I am making this suggestion.
- Also in Section 4 (“Laboratory assessments”), please consider providing a brief justification of why you chose to administer this specific product instead of a regular meal. Since the participants were not obese and, presumably, they consume regular food instead of these kinds of products.
- Please provide the manufacturer of catalogue 11885-084 (DMEM) listed in line 240.
- Lines 255-259 list the primers and PCR protocol used. Please specify if you designed the primers yourselves (include the software used if so) or provide a reference for the paper from which these were taken. Also, please specify the temperatures and times “recommended by Applied Biosystems”.
- In section “Western blot analysis” (lines 261-273), please specify centrifugation in g, and antibody dilutions used.
- In Figure 1, the graphs appear to be pixelated or with some artifact that makes the text look out of focus. Please consider providing a higher-quality version of these graphs.
- Finally, lines 149-151 mention the reported relationship between ZAG and lipid metabolism. As a suggestion for future works, please consider analyzing your participants’ serum lipids, including free fatty acids, since these may also provide you with some interesting information. Some anthropometric values like BMI, fat percentage, etc., could also be considered in order to explain the apparently contrasting data discussed in lines 156-165.
Author Response
Thank you very much for the revision process of our manuscript entitled “Acute downregulation of Zinc α2-glycoprotein by insulin in subjects without obesity” (Ms # ijms-3480660).
1.- Manuscript titled “Acute downregulation of Zinc α2-glycoprotein by insulin in subjects without obesity” reports in vitro and human-derived data regarding the effects of insulin and other stimuli on zinc α2-glycoprotein (ZAG). The data is interesting, although some aspects of the manuscript could be improved. The manuscript’s title suggests that the observed effects on ZAG were due to insulin in human subjects. This is slightly misleading since, strictly speaking, you administered insulin to HepG2 cells, but not to the participants. They were instead administered a glucose load, a macronutrient load or glucagon. Although human and in vitro data agree with each other, the stimuli are not the same on both models. Thus, please consider rephrasing the title to more adequately reflect your methods.
We agree with this comment, and therefore, the tittle has been changed to “Acute downregulation of Zinc α2-glycoprotein in response to metabolic stimuli: Evidence from Human and HepG2 Cell Studies”.
2.- Lines 65-70 mention that ZAG has been shown to decrease in obese individuals, and “suggesting a negative correlation with insulin resistance” (line 69). As currently written, the information suggests that obesity negatively correlates with ZAG, while insulin resistance is not specifically mentioned. Thus, please consider specifically mentioning that insulin resistance is present in the individuals of the cited papers, or state that ZAG negatively correlates with obesity, whichever is the most appropriate.
Following this comment, we have remarked that ZAG negatively correlates with adiposity.
3.- The abstract and Section 4 “Laboratory assessments” (and elsewhere in the manuscript) mention a “standard meal” or “standard food”, however, the “Fortisip” product administered appears to be more of a ready-to-drink product instead of a solid meal per se. Please consider providing a better description and/or nomenclature of the product in order to make its nature clear to the reader. Upon first reading your manuscript, I thought the meal used was an actual solid meal, and not a liquid product, which is why I am making this suggestion.
We thank the reviewer for their valuable comment and for highlighting this important point regarding the terminology used in our manuscript. We agree that the term 'standard meal' may be misleading, as the product administered was indeed a ready-to-drink nutritional supplement (Fortisip) rather than a solid meal. To improve clarity, we have revised the manuscript to refer to it as a 'standardized liquid nutritional supplement' instead of 'standard meal' or 'standard food.'
4.- Also in Section 4 (“Laboratory assessments”), please consider providing a brief justification of why you chose to administer this specific product instead of a regular meal. Since the participants were not obese and, presumably, they consume regular food instead of these kinds of products.
In Section 4 ('Laboratory assessments'), we have provided a justification for using this product. Fortisip was chosen due to its standardized macronutrient composition, ease of administration, and reproducibility across participants. This ensures consistency in nutrient delivery and minimizes variability that could arise from differences in the composition and absorption of regular solid meals. Moreover, despite participants being non-obese and presumably consuming regular food in their daily lives, using a standardized liquid product allows for better control of the experimental conditions and reduces potential confounding factors related to food texture, mastication, and digestion rates.
5.- Please provide the manufacturer of catalogue 11885-084 (DMEM) listed in line 240.
We thank the reviewer for this observation. We have included Thermo Fisher Scientific (Waltham, MA, USA) as the manufacturer of DMEM (catalogue number 11885-084) in the revised manuscript.
6.- Lines 255-259 list the primers and PCR protocol used. Please specify if you designed the primers yourselves (include the software used if so) or provide a reference for the paper from which these were taken. Also, please specify the temperatures and times “recommended by Applied Biosystems”.
The primers used for amplification were adopted from the study by Selva et al. (2009), which demonstrated their effectiveness under similar conditions. The corresponding reference has been included in the text. In addition, the PCR conditions were established according to the manufacturer's recommendations, including an initial denaturation at 95 °C for 10 minutes, followed by 40 cycles of denaturation at 95 °C for 15 seconds and annealing/extension at 60 °C for 1 minute.
7.- In section “Western blot analysis” (lines 261-273), please specify centrifugation in g, and antibody dilutions used.
We thank the reviewer for their valuable comment. We have included the requested information in the revised manuscript. Specifically, we have specified the centrifugation force (13,500 g) and the antibody dilutions used for Western blot analysis (1:1000 for ZAG and 1:10,000 for cyclophilin A) in the corresponding section.
8.- In Figure 1, the graphs appear to be pixelated or with some artifact that makes the text look out of focus. Please consider providing a higher-quality version of these graphs.
We thank the reviewer for their observation. Following their suggestion, we have provided a higher-quality version of Figure 1 to ensure better clarity and resolution of the graphs and text.
9.- Finally, lines 149-151 mention the reported relationship between ZAG and lipid metabolism. As a suggestion for future works, please consider analyzing your participants’ serum lipids, including free fatty acids, since these may also provide you with some interesting information. Some anthropometric values like BMI, fat percentage, etc., could also be considered in order to explain the apparently contrasting data discussed in lines 156-165.
We thank the reviewer for this insightful suggestion. We agree that analyzing serum lipids, including free fatty acids, as well as additional measures such as body fat percentage, could provide valuable information to further explain the observed data. We will certainly consider including them in future research to deepen our understanding of the relationship between ZAG, lipid metabolism, and body composition.
Round 2
Reviewer 2 Report
Comments and Suggestions for Authors
The results of study may somewhat contribute to the understanding of the acute effects of insulin on ZAG regulation in normal population. But the contribution is limited. However, if the subject is diabetes or obesity patient, the results will be interesting and important. Therefore, it is suggested that this manuscript perhaps can be accepted as Note or Letter to editor type.
Author Response
Thank you very much for the revision process of our manuscript entitled “Acute downregulation of Zinc α2-glycoprotein by insulin in subjects without obesity” (Ms # ijms-3480660).
1.- The results of study may somewhat contribute to the understanding of the acute effects of insulin on ZAG regulation in normal population. But the contribution is limited. However, if the subject is diabetes or obesity patient, the results will be interesting and important. Therefore, it is suggested that this manuscript perhaps can be accepted as Note or Letter to editor type.
We sincerely appreciate your evaluation and comments on our manuscript. We highly value your time and effort in reviewing our work.
We would like the article to continue being considered in its current format. However, if the journal deems it necessary to adapt it to the newly proposed structure, we are willing to make the required adjustments to align with their guidelines.
We remain available for any further indications and once again, thank you for your consideration.
Reviewer 3 Report
Comments and Suggestions for Authors
Manuscript titled “Acute downregulation of Zinc α2-glycoprotein in response to metabolic stimuli: Evidence from Human and HepG2 Cell Studies” reports in vitro and human-derived data regarding the effects of insulin and other stimuli on zinc α2-glycoprotein (ZAG). The present version of the document was modified according to comments and suggestions made during an initial review, those made by the present reviewer include:
- Rephrasing the title to better describe the authors’ methodological approach. The title was modified, and is now more descriptive of what the manuscript reports.
- In the introduction, rephrasing or providing a reference to better describe the relationship between ZAG and adiposity. The phrase was modified accordingly.
- Rephrasing or clarifying that the standard meal administered to the participants is a standardized ready-to-drink product, instead of a solid meal. The terminology used was modified throughout the document.
- Providing a brief justification for the use of the aforementioned product, instead of a regular solid meal. The authors have provided a justification for their choice of treatment.
- Providing manufacturer information for DMEM. The missing information was added.
- Specifying if the primers were designed by the authors or if they were taken from another paper; also including PCR temperatures. The authors have clarified the origin of the primers and provided a reference; PCR temperatures were added as well.
- Providing centrifugation data in g and antibody dilutions used. Both values have been included.
- Providing a clearer image for figure 1. The image was replaced for a clearer one.
According to the aforementioned changes made by the authors, it is apparent that they adequately considered and addressed all comments and suggestions. One final comment is that figure 1A shows “Insulina” instead of “Insulin”; although this should be fixed before final publication, it does not merit another revision.
Author Response
Thank you very much for the revision process of our manuscript entitled “Acute downregulation of Zinc α2-glycoprotein by insulin in subjects without obesity” (Ms # ijms-3480660).
1.- Manuscript titled “Acute downregulation of Zinc α2-glycoprotein in response to metabolic stimuli: Evidence from Human and HepG2 Cell Studies” reports in vitro and human-derived data regarding the effects of insulin and other stimuli on zinc α2-glycoprotein (ZAG). The present version of the document was modified according to comments and suggestions made during an initial review, those made by the present reviewer include:
- Rephrasing the title to better describe the authors’ methodological approach. The title was modified, and is now more descriptive of what the manuscript reports.
- In the introduction, rephrasing or providing a reference to better describe the relationship between ZAG and adiposity. The phrase was modified accordingly.
- Rephrasing or clarifying that the standard meal administered to the participants is a standardized ready-to-drink product, instead of a solid meal. The terminology used was modified throughout the document.
- Providing a brief justification for the use of the aforementioned product, instead of a regular solid meal. The authors have provided a justification for their choice of treatment.
- Providing manufacturer information for DMEM. The missing information was added.
- Specifying if the primers were designed by the authors or if they were taken from another paper; also including PCR temperatures. The authors have clarified the origin of the primers and provided a reference; PCR temperatures were added as well.
- Providing centrifugation data in g and antibody dilutions used. Both values have been included.
- Providing a clearer image for figure 1. The image was replaced for a clearer one.
According to the aforementioned changes made by the authors, it is apparent that they adequately considered and addressed all comments and suggestions. One final comment is that figure 1A shows “Insulina” instead of “Insulin”; although this should be fixed before final publication, it does not merit another revision.
I sincerely appreciate your thoughtful review of our manuscript, “Acute downregulation of Zinc α2-glycoprotein in response to metabolic stimuli: Evidence from Human and HepG2 Cell Studies”. Your detailed comments and suggestions have been invaluable in refining our work.
We are grateful for your positive assessment and for acknowledging the modifications made in response to the initial review. Regarding your final comment on Figure 1A, we will ensure that "Insulina" is corrected to "Insulin" before final publication.
We hope that the manuscript can proceed toward acceptance in its current form, but we remain open to any final editorial adjustments if necessary.
Thank you once again for your time and constructive feedback.